# Natural Antioxidants: A Novel Therapeutic Approach to Autism Spectrum Disorders?

**DOI:** 10.3390/antiox9121186

**Published:** 2020-11-26

**Authors:** Luca Pangrazzi, Luigi Balasco, Yuri Bozzi

**Affiliations:** Center for Mind/Brain Sciences (CIMeC), Neurodevelopmental Disorders Research Group, University of Trento, 38068 Rovereto, Italy; luigi.balasco@unitn.it

**Keywords:** autism, inflammation, oxidative stress

## Abstract

Autism spectrum disorders (ASD) are a group of neurodevelopmental syndromes with both genetic and environmental origins. Several recent studies have shown that inflammation and oxidative stress may play a key role in supporting the pathogenesis and the severity of ASD. Thus, the administration of anti-inflammatory and antioxidant molecules may represent a promising strategy to counteract pathological behaviors in ASD patients. In the current review, results from recent literature showing how natural antioxidants may be beneficial in the context of ASD will be discussed. Interestingly, many antioxidant molecules available in nature show anti-inflammatory activity. Thus, after introducing ASD and the role of the vitamin E/vitamin C/glutathione network in scavenging intracellular reactive oxygen species (ROS) and the impairments observed with ASD, we discuss the concept of functional food and nutraceutical compounds. Furthermore, the effects of well-known nutraceutical compounds on ASD individuals and animal models of ASD are summarized. Finally, the importance of nutraceutical compounds as support therapy useful in reducing the symptoms in autistic people is discussed.

## 1. Introduction

Autism spectrum disorders (ASD) represent a heterogeneous group of neurodevelopmental syndromes clinically characterized by deficits in social interaction/communication and repetitive/stereotyped behaviors, accompanied by a different degree of sensory deficits and other comorbidities [1]. The pathogenesis of ASD is complex and not fully disclosed. Research has identified both genetic and environmental factors, especially those involving fetal and early life development, in describing this complex condition.

Although studies performed in twins with ASD have revealed a high degree of heritability (38–54%), several meta-analyses studies have also shown that genetic-independent causes of ASD do exist [2]. In the last two decades, a consistent number of genetic modifications has been reported to increase the risk for developing ASD or autistic symptoms. Approximately 10% of ASD cases are of genetic etiology (Fragile X syndrome [3], tuberous sclerosis [4], Cowden syndrome [5], and Angelman syndrome [6]); of those, single genetic mutations account for only 1–2% of cases [7], with the majority of cases remaining idiopathic. Recent advances in genetics and genomics have allowed for the identification of an increasingly high number of ASD-associated gene variants, including chromosomal deletions or duplications (copy number variations, CNVs), de novo and familial mutations, and epigenetic in candidate as well as novel genes [8,9,10]. Many of these mutations and CNVs have been the subject of investigation in mouse models and some of them revealed to be promising for ASD research. Several genetic mutations in ASD have been associated with genes coding for proteins involved in synaptic functions, such as SH3 and multiple ankyrin repeat domains 3 (SHANK), contactin-associated protein-like (CNTNAP), neuroligin (NLGN), and neurexin (NRXN). Some examples of CNVs associated with ASD include chromosomal loci 15q11-q13, 16p11.2, and the ubiquitin protein ligase E3A (UBE3A) genes. Moreover, a subset of single gene mutations associated with ASD are also responsible for other neurodevelopmental disorders, including fragile X mental retardation syndrome 1 (FMR1) in fragile X syndrome, tuberous sclerosis 1(TSC1) in tuberous sclerosis, and methyl-CpG binding protein 2 (MECP2) in Rett syndrome [11]. There is a strong evidence indicating that highly convergent cellular mechanisms underlie the genetic and functional complexity of autism spectrum disorders. Indeed, most of ASD-associated genes code for proteins involved in synaptic signaling, transcriptional/post-transcriptional mechanisms and cell adhesion, and function of excitatory and inhibitory neurons. These mechanisms crucially operate during nervous system development, controlling the birth, differentiation, and migration of neuronal subtypes, along with the formation and maintenance of their connections [8,9]. Several studies have identified mutations and chromosomal aberrations in genes that are involved in brain development from intrauterine life through childhood. As frequent mutations in genes involved in the cytoarchitectural organization and neuronal connectivity have been found, ASD has been proposed to be a synaptopathy [12]. However, several lines of evidence suggest a role of inflammation in the underlying mechanism of ASD, despite some concern about whether it causes ASD onset or regulates ASD pathogenesis and symptomatology.

Immunity takes a key role in neural development of both central and peripheral nervous system, regulating several aspects such as neuronal proliferation, synapse formation, and plasticity, together with the removal of apoptotic neurons; in addition, immune dysfunction has also been shown to contribute to large number of neurological conditions [13,14,15]. Several studies over the last decades have highlighted altered immune responses in individuals with ASD [16,17]. People with ASD have increased levels of pro-inflammatory markers while having decreased levels of anti-inflammatory markers [18]. Moreover, we recently discussed how oxidative stress may support brain inflammation and ASD-like behaviors [19]. In particular, the levels of reactive oxygen species (ROS) scavenging enzymes, such as superoxide dismutases (SODs), catalase, glutathione peroxidase (GPx), and antioxidant molecule glutathione (GSH) are lower in individuals with ASD, in comparison with healthy controls. Moreover, several studies reported how neuroinflammatory conditions may be present in ASD patients, which are commonly linked and supported by high levels of ROS. Furthermore, post-mortem brain samples from ASD individuals revealed increased levels of pro-inflammatory markers [20] and increased microglia activation [21,22]. Given the current pieces of evidence, solutions that target oxidative stress and inflammation could represent a valid strategy to tackle ASD.

In this review, we summarize the recent literature about impairments in the vitamin-based antioxidant network in the context of ASD. In addition, we describe how natural antioxidants present in plants and food may help in counteracting ASD-like behaviors.

## 2. Evidence of Inflammatory State in Animal Models of ASD

Experimental animal models represent a fundamental element for the understanding of the etiology and pathogenesis of any disease and human condition, including ASD. Animal models can be considered as reliable if (a) they show the same diagnostic markers of a human condition, demonstrating resemblance to the human situation (face validity); (b) they show similarity with the causes of the disease (construct validity); and (c) they show expected responses to treatments that are effective in the human disease (predictive validity). This can be achieved in a relatively simple way whenever there are distinct markers for a disease. For example, in animal models of hemophilia, one has to show similar coagulation defects. However, it is more difficult to prove that an experimental animal is the suitable model for ASD. Indeed, according to the Diagnostic and Statistical Manual of Mental Disorders 5 (DSM-V), diagnosis of ASD is solely based on behavioral changes, and the behavior of a mouse is not the same as the behavior of a child or adult [1]. Despite this obstacle, a variety of neurobehavioral tests have been used to demonstrate a behavioral pattern that mimics the core behavior of ASD in animal models. Among typical behaviors assessed in mice and rats, directly related to the core symptoms of autism, are deficits in social interaction (i.e., three-chamber social approach paradigm), deficits in communication (i.e., analysis of ultrasonic vocalizations), increased repetitive/stereotyped motor behaviors (i.e., self-grooming or marble burying), and persistence on sameness and restricted interests (i.e., perseveration in the T maze or water maze) [23,24]. Moreover, rodents can also be tested for a number of other behaviors aiming at testing sensory deficits, which are reported to be present in a large subset of ASD subjects [25]. Despite the evident heterogeneity of the clinical manifestation in ASD, animal research allows consistent investigations of different aspects of the condition, from circuitry to cellular and molecular processes affected in autism.

ASD models can be grouped in three large categories: (i) induced by genetic perturbations, mimicking the genetic alterations found in humans with ASD; (ii) environmentally induced, by exposure of the pregnant animals to certain chemicals or infectious/inflammatory agents; and (iii) inbred strains, analogous to idiopathic cases of autism in which no genetic mutations have been identified. According to the Simons Foundation Autism Research Initiative (SFARI) gene database (http://gene.sfari.org/, as of November, 2020), there are up to now 284 genetic, 45 pharmacologically-induced, and 8 inbred mouse models of ASD.

The progress made in identifying genetic modification associated with ASD [8,9,10] has resulted in the generation of a multitude of mouse models by knock-out and knock-in mutations in ASD candidate genes. Thanks to these, it was possible to infer the effect of single mutations, allowing researchers to advance in the understanding of the biological bases underpinning this complex syndrome. Given the heterogeneity of genetic mutations (which can take place in different positions along the DNA, thus in different genes) and the resulting increasing number of genetic mouse models that have been generated, different animal models reproduce various aspects of ASD symptomatology, with the best animal model keeping together all three types of validity criteria.

Despite the advancement in genetic screenings of ASD populations, for the majority of cases the exact etiology of ASD remains unknown. Novel technologies and large population-based studies have provided new insight into the risk architecture of ASD and the possible role of environmental factors in its etiology. If earlier studies on twins suggested heritability as high as 80–90% for ASD with little contribution from the environment [26], according to recent evidence, up to 40–50% of variance in ASD liability is determined by environmental factors [27,28]. Among the most documented environmental risk factors are maternal infection [29] and prenatal exposure to the anticonvulsant drug valproic acid (VPA) [30]. To this first category belong mouse models of maternal respiratory infection with influenza virus and maternal immune activation (MIA) with either polyinosine/cytosine (poly(I:C), a synthetic, double-stranded RNA that evokes an antiviral-like immune reaction), or lipopolysaccharide (LPS, which evokes an antibacterial-like immune reaction) [31]. Poly (I:C) MIA offspring exhibit behaviors similar to the core symptoms of autism—deficits in social interaction and communication (USVs) and increased repetitive/stereotyped motor behaviors [32]. Whereas maternal LPS administration yields offspring with some of the same features, and a few of the abnormal behaviors can be reversed by antipsychotic drug treatment [33]. To the second category belongs instead the mouse model of prenatal exposure to VPA. A single injection of VPA in pregnant mice results in behavioral abnormalities relevant to the core symptoms of autism in the offspring [34,35].

In addition to the genetically modified and environmentally induced models of ASD, several inbred mouse strains also incorporate face validity as ASD models since they display robust and well-replicated social deficits and repetitive behaviors. Between the most popular inbred strains are A/J, BALB/cByJ (BALB), BTBR T+Itpr3tf/J (BTBR), C58/J (C58), and 129S1/SvImJ mice, which exhibit lack of sociability, as compared to inbred mouse strains with high sociability, such as C57BL/6J (B6) and FVB/NJ mice. Of these, BTBR has been the most extensively characterized and well replicated for ASD-related behaviors [36,37]. However, one might wonder how adequate the above models are. Indeed, in these models, only certain aspects of ASD are reproduced and we know, at the same time, that autism is a multifactorial condition. In animal models, we can observe manifestations associated with “human” autistic disorders, but it is difficult to identify whether these manifestations are the reasons for these disorders and their consequences, whether they are different consequences of identical reasons, or whether these manifestations are interconnected by some other relations. Despite this, we and many other researchers believe that the heterogeneity of face validity in mouse models of ASD could be converted from a problem into an advantage as long as investigators exploit these differences to advance our understanding of autism.

Nonetheless, certain aspects of ASD symptomatology such as inflammation and oxidative stress seem to be shared by many ASD mouse models [19]. Although the link between immune system alteration and ASD has long been postulated [38], only in the last decades has the research looked more closely at the potential contribution of the immune system to ASD. Studies performed in ASD patients indicated that immune system dysfunction is often supported by a strong inflammatory state [39,40]. Moreover, signs of microglia activation as well as increased inflammatory cytokines and chemokines (i.e., interferon (IFN)γ, interleukin (IL)-1β, IL-6, tumor necrosis factor (TNF), and chemokine C-C motif ligand (CCL)-2) have been found in the brains and cerebrospinal fluid of ASD subjects [20,41]. Immunological studies on mouse models of ASD have classically focused on MIA and VPA mouse models. Maternal immune activation (MIA) mouse offspring have been reported with altered immune functions showing systemic deficits in CD4^+^ TCRβ^+^ Foxp3^+^ CD25^+^ T regulatory cells and increased IL-6 and IL-17 cytokine production by CD4^+^ T cells [42]. Moreover, central neuro-inflammation and altered inflammatory responses, together with synaptic alterations, have been shown in VPA rodent models [43,44]. However, inflammatory alterations have also been found in genetic and inbred mouse models of ASD [19,45,46], suggesting that increased production of pro-inflammatory molecules in ASD may represent a strong contributor to the pathogenesis and the severity of these disorders. In the following chapters, we focus on natural antioxidants and their anti-inflammatory activity as potential target to treat ASD symptomatology.

## 3. Vitamin E/Vitamin C/Glutathione Antioxidant Network Controls Oxidative Stress

Vitamin E is present in lipid-rich plant products and vegetable oils, such as wheat germ, rice bran, coconut, soybean, and olives. Eight different isoforms have been identified: α-, β-, γ-, δ- tocopherols and α-, β-, γ-, δ-tocotrienols. Vitamin E is incorporated into cellular membranes, in which it effectively inhibits the peroxidation of lipids, as both tocopherols and tocotrienols scavenge highly reactive peroxyl radicals. Vitamin E does not work in isolation from other antioxidants, but it is part of an interlinking set of redox antioxidant cycles known as the “antioxidant network” [47]. When all vitamin E isoforms react against free radicals, they are converted into their oxidized tocopheroxyl/tocotrienoxyl forms. In order to be active again, vitamin E is converted from its free radical form into its reduced native state after the interaction between water- and lipid-soluble substances, using both enzymatic and non-enzymatic mechanisms.

Reduced vitamin C (dehidroascorbate, DHA) can directly regenerate vitamin E after being converted into ascorbate, its oxidized form. Thiol antioxidants such as glutathione can convert ascorbate into DHA, which is then available for regenerating again vitamin E. Thus, vitamin C is particularly effective at reducing oxidative damage when it is used in conjunction with vitamin E [48,49]. When vitamin E, vitamin C, and glutathione systems act synergistically, low steady-state concentrations of vitamin E radicals and ascorbate are present in the cells, and the loss or consumption of vitamins is prevented. Vitamin C is a potent antioxidant that can neutralize and remove oxidants, such as highly reactive molecules generated in metabolic processes in the brain but also in other organs [49]. Vitamin C plays a vital role in tissue growth, synthesis of vasoactive agents, immune regulation, and many other metabolic functions [50,51]. Decreased vitamin C levels are known to support the pathophysiology of various diseases, such as cancer, endocrinopathies, and neurological disorders [52]. The biological significance of vitamin C in the brain is related to the development of neurons, their functional maturation, and antioxidant responses. When a vitamin C deficiency is present, neurons show decreased growth and activity, as well as increased susceptibility to oxidative damage [53].

Another important molecule within the antioxidant network, necessary for the reconversion of ascorbate into DHA, is glutathione. Glutathione is a tripeptide (cysteine, glycine, and glutamic acid) present in high levels in most cells, which exists in two forms, reduced (GSH) and oxidized (GSSG). The ratio of GSH to GSSG determines the cell redox status of a cell. Resting healthy cells show a GSH/GSSG ratio > 100, while the ratio drops to 1 to 10 when cells are exposed to oxidant stress. De novo glutathione synthesis is mainly controlled by the cellular levels of the aminoacid cysteine, the availability of which is the rate-limiting step. For this reason, administration of antioxidant N-acetyl L-cysteine (NAC), a precursor of cysteine, is known to boost the synthesis of intracellular glutathione [54,55]. Thus, the integration of several antioxidants within the antioxidant network is fundamental for counteracting the onset of oxidative stress within the body, a condition known to support the pathogenesis of diseases.

## 4. Vitamin E/Vitamin C/Glutathione Network is Impaired with ASD

Studies have shown that all players involved in the vitamin E/vitamin C/glutathione network are impaired with ASD. Alterations in the expression of major antioxidant enzymes of ROS scavenging system are present in both ASD mouse models and autistic patients, in the brain but also in the peripheral blood [19,56,57,58,59,60,61]. This indicates that levels of oxidative stress may be elevated with ASD.

It has been described that concentration of vitamin E in the blood was reduced and was associated with ASD-like behaviors in autistic people [57,62]. Future studies must assess whether vitamin E supplementation may attenuate ASD-like symptoms. Vitamin C deficiency has additionally been observed with ASD. Differently from lipid-soluble vitamin E, vitamin C is not stored in any reservoir within the body. As it is not endogenously produced, it is rapidly lost within the urine if not supplemented from the diet. Thus, plasma concentration of vitamin C is related to its dietary intake and it has a half-life of 12–24 h [63]. The consumption of very low amounts of food rich in vitamin C may lead to scurvy, a rare disease that is mainly present in people with unusual eating habits, suffering of alcoholism, or with mental disorders. Recently, increased prevalence of scurvy has been observed in children with ASD [64]. This situation may be caused by vitamin C malabsorption and/or low vegetable and fruit intake. In addition, low vitamin C levels have been observed in the blood of children with ASD, in comparison with healthy controls [65,66]. Another aspect which can typically be observed in ASD children is a decreased level of glutathione, particularly in its reduced form [56,67,68,69]. In parallel, the levels of oxidized glutathione are higher than in normally developing children. As reduced glutathione is needed to regenerate DHA from ascorbate (Figure 1), lower levels of vitamin C in its reduced form can be expected with ASD. Consequently, increased amounts of vitamin E in its oxidized tocopheroxyl/tocotrienoxyl form may be present in the presence of ASD.

Clinical trials have been planned in order to counteract this situation, with the aim of restoring vitamin E/vitamin C/glutathione antioxidant network in ASD individuals [70]. In a double-blind, placebo-controlled study, the safety and efficacy of glutathione alone or glutathione, vitamin C, and NAC treatment was evaluated in children with autism [70]. A placebo or one of the two treatments was administered to the participants for 8 weeks. The authors observed an improvement in both developmental skills and behavior with both glutathione or the combined therapy glutathione, vitamin C, and NAC therapy as compared to placebo therapy. A positive correlation between changes in behavior and GSH/GSSG ratio was additionally observed [70]. Thus, evidence exists that ASD-like symptoms may be improved by targeting the vitamin E/vitamin C/glutathione network. Whether nutritional interventions aiming at increasing the levels of vitamin E and vitamin C, both present within foods, may help in restoring an optimal functionality of the antioxidant network have not investigated yet but it can be expected. Thus, a careful evaluation of nutritional status should be performed in autistic patients before planning more elaborate testing. Indeed, as ASD are a very heterogeneous group of diseases, differences in the nutritional status of autistic people may be present. This may be very helpful to plan nutritional interventions to counteract ASD-like symptoms.

## 5. Functional Food and Nutraceutical Compounds

The first and fundamental role of food is to provide nutrients (i.e., carbohydrates, proteins, fats, vitamins, and minerals) to the body, which are necessary for growth, maintenance, and wellness. In addition, non-nutrients such as fibers, probiotics, antioxidants, and bioactive molecules are included in the food. Although they are not directly required to support the basic functions of organisms, their presence is beneficial to promote health and wellness. When a kind of food can provide health benefits in addition to its nutritional value, it is defined as functional food. Generally, a typical hallmark of these aliments is the presence of antioxidant molecules such as carotenoids, flavonoids, and phenols. Typical examples of functional foods are many kinds of fruits, vegetables, nuts, seeds, whole grains, legumes, and seafood. When functional compounds are isolated from the food matrix, they are called nutraceuticals. This term was coined by De Felice in 1989, combining the words “nutrition” and “pharmaceutics”, to indicate “food or parts of food that provide medical or health benefits, including the prevention and treatment of diseases” [71]. In this way, when interesting molecules are identified within functional foods, they can potentially be enriched and sold as nutraceuticals on the market. Despite this, it is important to consider that bioactive compounds may be less effective when isolated from functional foods, and, in some cases, toxic effects may appear. Nevertheless, these molecules may be helpful in counteracting pathological conditions and supporting body health. Recently, many studies have started exploring the role of nutraceuticals in the context of chronic disorders such as cancer, diabetes, atherosclerosis, cardiovascular diseases, and neurological disorders, and interesting results have been achieved [72,73,74,75,76]. In addition, evidence exists that nutraceutical compounds may also play a beneficial role in individuals with ASD. In these studies, the antioxidant potential of nutraceutical compounds played a major role in improving the disease situation. Thus, functional food may represent a powerful tool for counteracting a broad spectrum of diseases thanks to the combination of several nutraceuticals, most of which are antioxidant compounds. As previously discussed, nutritional interventions may be important to counteract ASD-like symptoms in individuals with ASD. We next summarize recent studies describing how not only the antioxidant activity of vitamins (nutrients), but also how the antioxidant potential of functional foods/nutraceuticals (non-nutrients) may help in counteracting autistic-like behaviors.

## 6. Nutraceutical Compounds and ASD

### 6.1. Carotenoids

Carotenoids are a class of fat-soluble, red, orange, and yellow natural pigments synthesized by plants, bacteria, and fungi. These molecules present in and give the characteristic color to carrots, tomatoes, and pumpkins. Importantly, many carotenoids show antioxidant and anti-inflammatory properties [77]. Several studies showed that people consuming more carotenoids in their diets had reduced risk of chronic diseases, such as cardiovascular diseases, photosensitivity diseases, cataracts, and age-related macular degeneration [78]. In addition, carotenoids were shown to improve cognitive performance in middle-aged people, suggesting that these compounds can be helpful in supporting brain health [79]. The importance of carotenoids in the context of ASD has been described by the work of Krajcovicova-Kudlackova and colleagues, which reported that blood levels of lycopene, a type of carotenoid, were reduced in autistic patients [62].

Among all the carotenoids, astaxanthin, a carotenoid belonging to the family of terpenes, has shown promising effects in the context of neurodegenerative disorders. Like many carotenoids, astaxanthin is a lipid-soluble pigment with a red-orange color, and a strong capability of reducing reactive oxidizing molecules. Interestingly, astaxanthin has been shown to display the strongest antioxidant potential among all the carotenoids [80,81]. This compound is naturally produced by microalgae *Haematococcus pluvialis* and the yeast fungus *Xanthophyllomyces dendrorhous*. In particular, when the algae are stressed by increased salinity, lack of nutrients, or excessive sunshine, they create astaxanthin. Animals who feed on the algae, such as red trout, salmon, flamingos, and crustaceans (i.e., shrimps, crabs, and lobsters), subsequently acquire the red-orange astaxanthin pigmentation. Thus, a nutrition based on algae and red-colored seafood leads to a consistent astaxanthin intake. Several studies performed in the context of cardiovascular diseases, metabolic syndromes, and cancer have described that astaxanthin shows a strong anti-inflammatory potential [82]. More recently, it was documented that this molecule may additionally show a neuroprotective potential, particularly in the elderly and in the presence of neurodegeneration, as it displays a therapeutic role in preserving cognitive function [83]. In addition, astaxanthin may attenuate microglial activation and the release of pro-inflammatory cytokines. Furthermore, this molecule is known to protect neuronal integrity [84,85,86]. In a prenatal valproic acid (VPA)-induced mouse model of ASD, astaxanthin was shown to improve behavioral disorders and oxidative stress in the treated animals [87]. Thus, if these results are to be confirmed in humans, the consumption of astaxanthin-rich food may be proposed to autistic patients in order to attenuate ASD-like symptoms.

### 6.2. Polyphenols

In addition to carotenoids, polyphenols have been shown to play a beneficial role in ASD. Phenols are a class of nutraceutical compounds representing the most abundant group of secondary metabolites produced by plants. These molecules are spread throughout many foods, such as vegetables, fruits, and seeds. As they contain one or more hydroxyl groups attached to a benzene ring, they have the capability of donating electrons to oxidized molecules. For this reason, phenols are considered strong antioxidant molecules. Phenolic compounds are a very heterogeneous group, and they show different chemical properties. Several polyphenols have been shown to modulate sirtuin 1 signaling pathway and regulate key cell processes, such as cell cycle, DNA repair, protein aggregation, inflammation, and mitochondrial function [88]. In addition, this class of molecules can also interfere with other key cell signaling pathways (such as those controlled by activator protein 1 (AP-1), nuclear factor kappa-light-chain-enhancer of activated B cells (NF-kB), and signal transducer and activator of transcription 1 (STAT1), ultimately leading to protection against oxidative stress, inflammation, and cell proliferation [89,90]. Among the polyphenol class are flavonoids, which include quercetin and luteolin, both showing antioxidant and anti-inflammatory properties [91,92].

Recent studies have shown that dietary polyphenols can be promising molecules to alleviate ASD symptoms. Several studies reported that some polyphenols, including resveratrol, regulate mitochondrial activity and prevent mitochondrial dysfunction, typically present in individuals with ASD [93]. Furthermore, resveratrol was efficient in preventing the downregulation of cytochrome c oxidase and in suppressing the production of ROS within mitochondria in azide-stimulated primary hippocampal astrocytes [94]. It is important to consider that polyphenols may have low bioavailability within the brain, as not all of these compounds can cross the blood–brain barrier. Despite this, they are present at high concentrations in the intestinal lumen [95,96], and therefore they may modulate the microbiota, which is of great importance for ASD typically characterized by impairments in the microbiota–gut–brain axis [97]. To date, very few studies have investigated the potential benefits of dietary polyphenols in ASD patients.

Taliou and co-workers performed a pilot study including 50 children diagnosed with ASD [98]. An oral formulation including a combination of flavonoids (quercetin, luteolin, and quercetin glycoside rutin) was administered daily for 26 weeks. The results showed that ASD children receiving the polyphenols showed a significant improvement in communication and concentration, with reduced abnormal behaviors. More recently, it was shown that children treated with the same polyphenolic formulation improved autistic-like behaviors and, in parallel, the serum levels of pro-inflammatory cytokines IL-6 and TNF were significantly reduced when compared to the beginning of the study [99]. Thus, the administration of polyphenols may lead to decreased neuroinflammation, promoting an improvement in ASD-related symptoms in the children as a result. In addition, a single case report described that a 10-year-old ASD child who received microgranules including palmitoylethanolamide and luteolin twice a day for one year showed an improvement in ASD-like behaviors [100]. Similar effects were observed in VPA-induced autistic mice after the administration of the same formulation [100]. Taken together, evidence exists that polyphenols may represent a class of natural molecules important for counteracting ASD symptoms.

### 6.3. Omega-3

Omega-3, known also as n-3 or short-chain fatty acids, are a class of polyunsaturated fatty acids (PUFAs) extensively distributed in nature, very important constituent of lipid metabolism. Three main types of omega-3 fatty acids are fundamental for human physiology: α-linolenic acid (ALA), present in plant oils, and eicosapentaenoic acid (EPA) and docosahexaenoic acid (DHA), both commonly found in marine oils. The main sources of plant oils containing ALA are represented by seeds, walnut, and clary sage seed oil, while animal omega−3 fatty acids EPA and DHA are mostly included in fish, fish oils, and chicken eggs. As mammals are unable to synthesize ALA omega-3, this compound must be introduced through the diet. In addition, ALA is required for the synthesis of EPA and DHA, and thus deficits in the intake of ALA lead to an overall omega-3 reduction within the body. In contrast to long-chain, n-6 polyunsaturated fatty acids (n-6 PUFAs), which support inflammatory processes, omega-3 are anti-inflammatory molecules and thus they are fundamental for keeping inflammation under control [101].

Impaired fatty acid metabolism is known to affect normal brain function. Indeed, omega-3 play an important role in the structure of the neuronal cell membranes, as well as in the development of myelin sheath [102]. In particular, DHA constitutes 90% of the whole amount of omega-3 in the human brain and 10–20% of total lipids [103]. A link between abnormal fatty acid metabolism and the pathophysiology of ASD has been reported [69,104]. Increased DHA intake reduces the risk of bipolar disorder, schizophrenia, and depression, while low levels of this molecule are a potential risk factor for mental illness [105]. In keeping with these findings, children with ASD showed lower DHA and EPA levels and lower total omega-3 serum levels compared to healthy, age-matched children [106]. This condition may be caused by defects in enzymes involved in the DHA and EPA production from ALA, impairment in cell membrane incorporation, or dysfunction in mitochondrial PUFA oxidation [102,107]. More recently, a model linking ASD phenotype and DHA deficiency has been proposed [108]. The gut–brain axis was recently suggested as an alternative pathway for the omega-3 action against ASD. Indeed, n-3 fatty acid deficiency during perinatal period is known to alter intestinal microbial balance in offspring, leading to a reduction in bacterial density and a reduced proportion of Firmicutes to Bacteroidetes [109,110]. Omega-3 supplementation could increase the Lactobacillus and Bifidobacterium species in the gut and decrease the levels of potential pathobionts belonging to the Enterobacteriaceae family [109]. In addition, microbial overgrowth impaired the uptake and metabolism of PUFAs and other molecules. Taken together, omega-3 administration may improve ASD-like behaviors in autistic people, most-likely by reducing neuroinflammatory conditions within the brain and supporting the optimal functionality of gut–brain axis.

### 6.4. Plants as Sources of Nutraceutical Compounds to Treat ASD

Several compounds present in plants have been described to reduce brain inflammation and oxidative stress, and therefore these molecules may be very beneficial to treat ASD-like disorders [111]. Green tea (*Camellia sinensis*) extracts are an important source of polyphenols, such as flavonoids such as myricetin and quercetin [112]. When green tea extracts were daily administered to the VPA-treated mice from postnatal day 14 to day 40, the researchers documented an improvement of ASD-like behaviors [113]. The extracts were shown to reduce the expression of pro-inflammatory molecules TNF and C-reactive protein (CRP), and thus displayed evident anti-inflammatory properties [114]. One important ASD-related feature is the loss of Purkinje cell integrity within the cerebellum [115]. Importantly, treatment with 300 mg/kg green tea extracts lead to the regeneration of Purkinje cell layer in mice, suggesting that molecules present in the extract may potentially help in treating ASD [113].

The plants *Piper longum* and *Piper nigrum* (black pepper) are sources of another nutraceutical compound, piperine, a well-known anxiolytic, neuroprotective, antioxidant molecule [116,117]. In addition, piperine is used to treat seizures disorders and enhances memory and cognition [118]. In the VPA mice treated with 20 mg/kg piperine, antioxidant markers and behavioral abnormalities could be rescued in comparison to untreated mice [119]. Clinical trials have been started to assess whether piperine may additionally be beneficial for ASD children [118]. Another plant with well-known neuroprotective effects against neuropsychiatric disorders is turmeric (*Curcuma longa*), whose properties can be mainly attributed to curcumin. This molecule can target several signaling pathways, and thus the most important effects of curcumin include the increase of intracellular glutathione levels, and reduction of mitochondrial dysfunction, inflammation, oxidative stress, and protein aggregation [120]. In a VPA-induced rat model, a single curcumin supplement increased the expression of antioxidant enzymes [121]. Furthermore, rats that received curcumin showed reduced levels of pro-inflammatory cytokines IL-6 and IFNγ. In another study performed in old rats, increased synaptic plasticity after regular curcumin intake has been observed [122]. Future clinical studies must investigate the effects of curcumin on individuals with ASD. The standardized extract of *Ginkgo biloba* leaves, known as EGb 761 (*Ginkgo biloba* extract 761), includes 24% flavone glycosides (quercetin, kaempferol, and isorhamnetin), 6% terpene lactones (2.6–3.2% bilobalide, and 2.8–3.4% ginkgolides A, B, and C), 0.8% ginkgolide B, and 3% bilobalide [123]. Many of the substances present in the extract were associated with neuroprotective effects in disorders such as Parkinson’s disease, Alzheimer’s disease, and ischemic stroke [124]. In an observational study performed in individuals with ASD, 100mg/kg *Ginkgo biloba* extract taken twice a day improved autistic behaviors such as hyperactivity, irritability, and inappropriate speech [125]. Future study should further investigate the potential benefits of Ginkgo biloba extract on larger cohorts of autistic people.

## 7. Conclusions

It is now becoming evident how natural antioxidant and anti-inflammatory molecules may consistently help in counteracting symptoms in people with ASD (as summarized in Table 1). These effects can potentially be achieved by both the administration of selected antioxidants such as vitamin E, vitamin C, or NAC, or after the consumption of food rich in nutraceutical compounds, which may show additive or synergistic effects. In particular, nutraceutical compounds are present at very high concentrations in plants, fish, and crustaceans, which therefore can be considered important functional foods. Importantly, these molecules are extremely helpful for body health, as they can target inflammation and oxidative stress, and decrease damage within several organs, including the brain. Furthermore, as ASD individuals are characterized not only by high levels of oxidative stress and pro-inflammatory molecules but also by nutritional deficits, it is therefore important to perform a nutritional evaluation of every patient. In this way, the most appropriate strategy of intervention to counteract ASD-like behaviors can rapidly be selected.

## Figures and Tables

**Figure 1 antioxidants-09-01186-f001:**
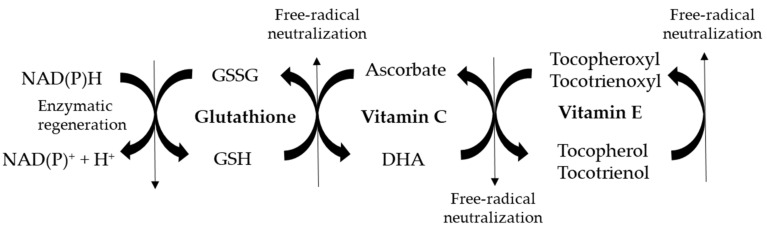
Schematic representation of vitamin E/vitamin C/glutathione antioxidant network involved in free-radical neutralization. GSSG: glutathione disulphide (oxidized glutathione); GSH: reduced glutathione; DHA: dehidroascorbate (reduced vitamin C).

**Table 1 antioxidants-09-01186-t001:** Summary of the studies describing the effects of natural antioxidants on autism spectrum disorder (ASD) animal models and humans with ASD. Dose, duration of the treatment, species, molecular effects, and behavioral improvements are reported in the table. IV: intravenously; IP: intraperitoneally; VPA: valproic acid.

Molecule/Formulation	Dose	Duration	Species	Molecular Effect	Behavioral Improvements	Reference
Glutathione	600 mg IV	once a week for 8 weeks	human	increased GSH/GSSG ratio	developmental skills and learning	70
Glutathione, vitamin C, NAC	600 mg, 2000 mg, 20 mg/kg(all IV)	once a week for 8 weeks	human	increased GSH/GSSG ratio	developmental skills and learning	70
Astaxanthin	2 mg/kg IP	once a day for 4 weeks	VPA-treated mouse	reduced oxidative stress (brain and liver)	sociability, social novelty and reduced anxiety	87
Capsule including: two flavonoids, luteolin, quercetin, rutin	in each capsule:>95% pure, 100 mg, 70 mg, 30 mg	1 capsule per 10 kg weight per day for 26 weeks	human	reduced IL-6 and TNF in the serum	communication, concentration, aberrant behavior	87,88
Palmitoylethanolamide and luteolin	700 mg orally	once a day for 1 year	human	--	stereotyped behaviors	100
Palmitoylethanolamide and luteolin	1 mg/kg orally	once a day for 3 months	VPA-treated mouse	reduced IL-6, IL-1β, Bax, and BCL-2 in the brain	social behaviors	100
DHA and EPA	800 mg, 700 mg orally	once a day for 6 weeks	human	--	reduced hyperactivity and stereotyped behaviors	105
DHA and EPA	1.5 g orally	once a day for 6 months	human	--	--	105
Green tea	300 mg/kg orally	once a day for 26 days	VPA-treated mouse	reduced lipid peroxidation and cerebellar damage	exploratory activity and spatial learning, reduced anxiety	113
Piperine	20 mg/kg IP	once a day for 26 days	VPA-treated mouse	increased GSH, decreased total nitrite and MDA	locomotor activity, social behaviors, spatial learning, reduced anxiety	119
Curcumin	1 g/kg IP	single dose	VPA-treated rat	reduced IL-6, IFNγ, glutamine and oxidative stress	--	121
Ginkgo biloba extract 761	100 mg/kg orally	twice a day for 4 weeks	human	--	aberrant behavior	125

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
