# Peer review of "Natural Antioxidants: A Novel Therapeutic Approach to Autism Spectrum Disorders?"

_antioxidants, 2020, doi:10.3390/antiox9121186_

Round 1

Reviewer 1 Report

The review by Pangrazzi et al. deals with a topic of considerable interest: the role of antioxidant compounds in autism spectrum disorders, with a particular attention to their therapeutic potential. The review is well structured and reasoned.

However, some modifications (listed below) are required to improve the quality of the manuscript:

  • check correct writing terms, sometimes they are written in American English sometimes in British English
  • some references cited, concerning topics on which many scientific studies have been carried out in recent years, are very old (about 30 years ago); add more recent references to support the concept
  • some statements lack appropriate reference (for example, see sentence on line 86: "Decreased vitamin C levels are known to support the pathophysiology of various diseases, such as cancer, endocrinopathies, and neurological disorders.”)
  • some concepts are supported by the work of many scientific studies and not only by the work published by the research group presenting this manuscript (for example, see line 105: "Alterations in the expression of major antioxidants ...."); as this is a review, the other most significant references on the subject should be added
  • in the manuscript there are some minor grammatical errors:

line 156: instead of “…isolated form functional food…” correct with “…isolated from functional food…”

line 185: “This compound is naturally produced in by…” delete “in”

Author Response

The review by Pangrazzi et al. deals with a topic of considerable interest: the role of antioxidant compounds in autism spectrum disorders, with a particular attention to their therapeutic potential. The review is well structured and reasoned.

However, some modifications (listed below) are required to improve the quality of the manuscript:

  • check correct writing terms, sometimes they are written in American English sometimes in British English

Writing terms have been corrected from British to American English (i.e. “behavioural” corrected with “behavioral” and “signalling” with “signaling”)

  • some references cited, concerning topics on which many scientific studies have been carried out in recent years, are very old (about 30 years ago); add more recent references to support the concept

In line with the suggestion of the reviewer, three more recent references have been added (REF.50, REF 53 and REF.84). In addition, one old reference has been replaced with a more recent one (REF.52)

  • some statements lack appropriate reference (for example, see sentence on line 86: "Decreased vitamin C levels are known to support the pathophysiology of various diseases, such as cancer, endocrinopathies, and neurological disorders.”)

A new reference to support this statement (REF.53) has now been added.

  • some concepts are supported by the work of many scientific studies and not only by the work published by the research group presenting this manuscript (for example, see line 105: "Alterations in the expression of major antioxidants ...."); as this is a review, the other most significant references on the subject should be added.

We agree with the reviewer and we now added new references about the work of research groups describing alterations in the expression of antioxidant enzymes. The old reference (REF.19) has been kept as a last summary reference.

  • in the manuscript there are some minor grammatical errors:

line 156: instead of “…isolated form functional food…” correct with “…isolated from functional food…”

line 185: “This compound is naturally produced in by…” delete “in”

Grammatical errors have been corrected.

Reviewer 2 Report

The present review article demonstrated the effects of nutraceutical compounds on ASD.

The title was "Role of natural antioxidants in autism spectrum disorders" but the number of references studying the direct effects of natural antioxidants on ASD might be small and insufficient.

Section 4 appeared twice. Check and correct.

Line 158-160, "many studies have started.....[39-43]"; the references 39-43 might be same laboratory or research group shown as same authors. Not one but some different research groups should be referred.

Author Response

The present review article demonstrated the effects of nutraceutical compounds on ASD.

The title was "Role of natural antioxidants in autism spectrum disorders" but the number of references studying the direct effects of natural antioxidants on ASD might be small and insufficient.

The role of natural antioxidants on ASD has only started being investigated in the last years. For this reason, at least for some antioxidant molecules, their effects on ASD-like behaviors are unknown. Despite this, for each molecule, we always included a paragraph and a summary table describing its known effects on ASD, in mouse models and/or humans. To better clarify that the topic that we addressed is speculative, we changed the title in “Natural antioxidants: a novel therapeutic approach to autism spectrum disorders?”

Section 4 appeared twice. Check and correct.

The mistake has been corrected.

Line 158-160, "many studies have started.....[39-43]"; the references 39-43 might be same laboratory or research group shown as same authors. Not one but some different research groups should be referred.

We agree with the reviewer and we added new references from different research groups (REF.75 and REF.76)